# Comparative Study Regarding the Adherence to the Mediterranean Diet and the Eating Habits of Two Groups—The Romanian Children and Adolescents Living in Nord-West of Romania and Their Romanian Counterparts Living in Italy

**DOI:** 10.3390/foods10092045

**Published:** 2021-08-31

**Authors:** Costanza Pira, Gianfranco Trapani, Maurizio Fadda, Concetta Finocchiaro, Enrico Bertino, Alessandra Coscia, Catalina Ciocan, Magdalena Cuciureanu, Simona-Codruţa Hegheş, Maria Vranceanu, Doina Miere, Lorena Filip

**Affiliations:** 1Department of Dietetics and Clinical Nutrition at A.O.U. Città della Salute e della Scienza in Turin, 10100 Turin, Italy; mfadda@cittadellasalute.to.it (M.F.); ettafinocchiaro@gmail.com (C.F.); 2Alfred Nobel Friend’s Studies Center, 18030 Sanremo, Italy; gianfranco@trapanigianfranco.it; 3Neonatal Unit of Turin University at A.O.U. Città della Salute e della Scienza, 10100 Turin, Italy; enrico.bertino@unito.it (E.B.); alessandra.coscia@unito.it (A.C.); 4Department of Occupational Medicine at A.O.U. Città della Salute e della Scienza in Turin, 10100 Turin, Italy; catalina.ciocan@unito.it; 5Departament of Pharmacology, Gr.T. Popa University of Medicine and Pharmacy, 700115 Iasi, Romania; 6Departament of Drug Analysis, “Iuliu Hatieganu” University of Medicine and Pharmacy, 400012 Cluj Napoca, Romania; cmaier@umfcluj.ro; 7Departament of Toxicology, “Iuliu Hatieganu” University of Medicine and Pharmacy, 400012 Cluj Napoca, Romania; marievranceanu@gmail.com; 8Departament of Bromatology, Hygiene, Nutrition, “Iuliu Hatieganu” University of Medicine and Pharmacy, 400012 Cluj Napoca, Romania; dmiere@umfcluj.ro (D.M.); lfilip@umfcluj.ro (L.F.)

**Keywords:** Mediterranean diet (MD), Western diet, immigration, Italy, Romania

## Abstract

Background: The Mediterranean diet (MD) is associated with significant health benefits, including prevention of noncommunicable diseases (NCDs). Given the important migratory flow from Romania to Italy in recent decades, this study seeks to evaluate the differences between the nutritional habits of Romanian children and adolescents in Romania compared with those of Romanian children who moved to Italy or were born in Italy from both Romanian parents. Method: To assess adherence to MD, parents of Romanian children in Romania (RCR) and Romanian children in Italy (RCI) answered questions from an adapted version of the KIDMED test. Results: The results show that the high KIDMED index among RCI is significantly higher than the same index among RCR (68.09 versus 17.76, *p* < 0.05). RCR obtained a higher KIDMED score on different items: they had a lower consumption of fast food and sweets but an increased consumption of nuts, yogurts, and cheese. Conclusions: RCI have a better adherence to MD, but, at the same time, they are more exposed to westernized diet and practice less physical activity. Nutrition education is an important tool for improving health outcome.

## 1. Introduction

The Mediterranean diet (MD) describes the traditional dietary habits of people living around the Mediterranean Sea, in the olive tree growing areas. MD was ascribed to the list of Intangible Cultural Heritage of UNESCO in November 2010 [1]: it represents not only a nutritional pattern, but also a lifestyle, which includes regional products and recipes, healthy ways of cooking (without overshadowing the pleasure of enjoying tasty dishes [2]), individual and group physical activity. Furthermore, MD has been associated with greater control of body mass as compared with some other diets [3,4].

MD is characterized by the prevalent consumption of fruits, vegetables, whole grain cereals, legumes, nuts, and seeds, with olive oil as the main source of added fat. Moreover, MD includes regular but moderate intake of fish and poultry meat (≥2 servings per week), daily low-fat dairy products (milk, yoghurt, and cheese), less than two servings per week of red meat, processed meat, and moderate wine drinking (in adulthood) [2,5,6]. Another important feature of this lifestyle is moderate to high physical activity (PA). Concerning nutrients, MD is characterized mainly by a high intake of low glycemic index carbohydrates, monounsaturated fatty acids (contained mainly in extra virgin olive oil—EVOO), dietary fibers, and antioxidants. MD comprises both the consumption of vegetable proteins and a balanced ratio of *n*-6 to *n*-3 fatty acids. In addition, MD plays an important role in the prevention of non-communicable diseases (NCDs) [7,8].

It is generally thought that the declining adherence to MD in Mediterranean countries over the last decades, in parallel with a shift towards a Western dietary pattern, may be associated with the incidence of NCDs, from childhood onward [9]. “Western diet” is richer in red and processed meat, butter, candies and sweets, fried foods saturated fat, refined grains, simple carbohydrates, and processed foods, associated with a sedentary behavior. This phenomenon has been named Nutrition Transition and is one of the factors involved in the high prevalence of overweight and obesity in Mediterranean Countries [10].

Among the factors that contribute to those changes, it is worth mentioning the diminished time and attention devoted to food acquisition and preparation, mental health issues (anxiety and depression) [11], the influence of social media [12], education, ethnicity [13], and parental socio-economic status [14].

Giving the fact that eating constitutes a key medium for social inclusion and intercultural dialogue [15], our question was, what changes of dietary patterns are associated to immigration. Romanian immigration to Italy started in the early’ 90s [16], and Romanian immigrants represent today around 21.0% of the total foreign population in Italy [17]. Even though there are few studies drawing the attention to the correlation between obesity and migration [18], there are no data regarding Romanian population.

The aim of this study was to compare the nutritional and physical habits of Romanian children in Italy (RCI)—descendants of Romanian immigrants—to those of Romanian children in Romania (RCR). Other purposes of the study were to assess to what degree these populations (from rural and urban areas) follow the MD and whether MD has an impact on children’s body mass index (BMI).

## 2. Materials and Methods

This cross-sectional study was conducted in Italy and Romania from July to December 2018. The study population comprised Romanian children and adolescents aged 1–16 years, living in Cluj, North–West region of Romania, and in Italy (Liguria, Piedmont, Calabria, Lombardy, and Emilia Romagna). Both parents of each child were ethnic Romanian. Participants were recruited during a routine visit to a primary care pediatrician (PCP) or a consultation with a nutritionist (in Romania).

### 2.1. Weight and Height Assessment

The participants were weighed with minimal clothing using a high-precision scale, and their height was measured with an altimeter (1 mm accuracy). BMI was calculated as the ratio between weight (kg) and squared height (m^2^). This study did not aim to assess adherence to the MD based on weight status quantified on percentiles. BMI can be an important indicator of healthy growth and development and enables comparison between children of the same sex and age.

### 2.2. KIDMED Test

To assess the adherence to MD between the two groups, we used an adapted and translated version of the KIDMED Test (Mediterranean diet quality index for children and teenagers) [19,20]. The KIDMED test is a tool to evaluate the adherence to the MD for children and youths. It was developed and validated by Serra-Majem et al. [19]. The translations to Romanian and Italian were carried out by the authors of the study. The anonymous questionnaire was completed by parents without supervision of the PCP or nutritionist (Table 1).

The index is based on a 16-question test that explores the consumption of fruits, vegetables, fish, pasta/rice, cereals, yoghurt/cheese/dairy products, nuts, commercial baked and processed foods, breakfast habits and the frequency of skipping breakfast, fast food intake frequency, sweets consumption, and olive oil use during meal at home. Questions denoting a negative connotation with respect to the MD were assigned a value of −1 (4 questions), and those with a positive one a value of +1 (12 questions). The overall score could range from 0 to 12. The sum obtained by adding the value of each question each was classified into three levels:(1)>8: optimal Mediterranean diet;(2)4–7: improvement needed to adjust intake to Mediterranean patterns;(3)≤3: extremely low diet quality [20].

### 2.3. Statistical Analysis

The normality of the analyzed samples was evaluated with the Shapiro–Wilk test, and the results were expressed as means ± standard deviation. Student’s independent *t*-test and Mann–Whitney U-test were used for normal distributed variables. Two-tailed chi-square or Fisher exact test was used to evaluate differences in categorical variables, as appropriate. A value of *p* < 0.05 was considered significant. All statistical analyses were carried out using the software package XLSTAT, version 2016.03.31199.

### 2.4. Ethical Approval

This cross-sectional study was conducted in accordance with the Declaration of Helsinki. Ethical approval for the study was obtained from the Research Ethics Committee (11/NW/0409) and Iuliu Hatieganu University of Medicine and Pharmacy Cluj-Napoca ethics committee (196/19 April 2018). All subjects gave their informed consent for inclusion before they participated in the study. 

## 3. Results

A total of 530 participants enrolled in the study, 414 RCR and 116 RCI, aged 1–16, with balanced distribution between sexes (Table 2).

Regarding the place of residence, most of the RCR and RCI lived in large towns (more than 60,000 inhabitants) and villages (less than 5000 inhabitants) (large towns, 41.37% RCI versus 34.54% RCR; small towns, 8.62% RCI versus 11.11% RCR; villages, 49.13% RCI versus 45.89% RCR; countryside: 0.86% RCI versus 8.45% RCR). Most RCI had been living in Italy for more than 5 years (53.44%) and 18.10% for 4 or 5 years (data not shown).

We assessed the parents’ employment status (1 = craftsman; 2 = domestic worker; 3 = commercial worker; 4 = unemployed, 5 = employee; 6 = freelancer; 7 = factory worker), and our results indicated that in most cases, mothers of RCR were employed (32.6%), while mothers of RCI were either unemployed (39.65%) or domestic workers (27.58%). More than half of fathers of RCR were artisans (21.74%) or clerks (31.64%), while 54.31% of fathers of RCR were laborers (Figure 1).

It is noteworthy that 43.71% of RCR allocated more than 3 h per week to physical activities as compared with 14.65% of RCI (Figure 2). Most of RCI (33.62%) did not have the possibility to practice physical activities (Figure 2).

Among RCR, the main seasoning was olive oil (39%), while among RCI, there was an increased usage of EVOO (33.54%). Both populations used butter, lard, seeds and olive oil (37.84% in RCI versus 34.56% in RCR) (Figure 3).

Regarding the KIDMED’s questions, it is worth mentioning that there were statistically significant differences between the RCI and RCR groups. RCI consumed more fruits, vegetables, fish, olive oil but also more pasta/rice, fast food, pastries, and sweets as compared with RCR. RCR had an increased daily consumption of yoghurts and/or cheese (40 g) compared with RCI (Table 3).

Even though there was no statistical difference between groups regarding the average KIDMED index (5.48 in RCI and 5.39 in RCR), most of RCR had a medium KIDMED index (63.12%) versus 12.93% in RCI (*p* < 0.05), while a large number of participants from the RCI group presented high KIDMED (68.09%) index relative to RCR (17.76%) (*p* < 0.05) (Figure 4).

## 4. Discussion

The MD is characterized by a high amount of polyphenols, dietary fiber, and ω-3 polyunsaturated fatty acids. This complex mixture of nutrients plays an important role both in the prevention of NCDs and in providing a favorable gut microbiota profile (Figure 5) [21,22]. The link between the antioxidant effect of polyphenols and the decrease in the incidence of cardiovascular diseases, metabolic syndrome, obesity, diabetes, as well as the positive effect on mental and reproductive health is well known. Due to the high production of short chain fatty acids (butyrate species), induced by MD, the subjects who follow the diet seem to have a lower incidence and risk of progression for several intestinal diseases, some types of cancer as well as the reduction of the cardio-metabolic pathologies [21,23,24].

Most of the studies evaluating adherence to MD among children have been carried out in Greece or Spain. There were some studies conducted in Italy as well [26,27,28,29], but none have evaluated MD adherence between RCI and RCR so far. In Italy, the Romanian community is very well represented; on 31 December 2018, Romanian residents in Italy numbered 1,190,091, confirming its position as the leading foreign community [30]. The aim of this study was to make an initial assessment of changes in the dietary habits between the RCI and RCR population. 

Our data showed that RCI acquired some typical habits of the Italian population: RCI had a higher score in some items typical of MD (Table 3). In fact, they ate more frequently a second fruit per day, they ate raw or cooked vegetables more than once a day, they consumed fish regularly (at least 2–3 times/week) and pasta or rice almost every day; they also used olive oil as seasoning at home. On the other hand, RCR reached a higher KIDMED score on other items: they went less frequently to fast food restaurants, they ate less sweets or candies during the week, and they consumed fewer pastries or other similar foods for breakfast. They also maintained some traditional Romanian habits, like regular consumption of nuts, yoghurts, and cheese, habits that are less well preserved by RCI.

Food consumption is affected by different factors, including food availability, accessibility, and choices, which in turn may be influenced by geography, demography, socioeconomic status, urbanization, globalization, marketing, and consumers [31]. Romanian families who came to Italy improved their social status and have greater and easier access to certain typical Italian foods (fruits, fish, olive oil) that are cheaper there than in their country of origin. At the same time, the globalization of fast food is beginning to influence the eating patterns of children in several countries, including Italian-native children and RCI. It is possible that for many RCI families, gathering in fast food restaurants is also a way of socializing and spending leisure time at weekends. Westernized lifestyle is also seen in Spain and Greece [32,33]. 

Given the fact that younger people are fast-food consumers and the role of grandparents in transmission of healthy eating habits in not always possible for the RCI population, school menus and education regarding healthy lifestyle in schools become increasingly important [34].

Some previous studies had found a correlation between MD and a lower BMI [35,36,37,38]. In the study conducted by Tognon et al. [39], high frequency-based Mediterranean diet score (fMDS) at baseline protected against increases in BMI, waist circumference, and waist-to-height ratio. In our research, there was no difference between BMI of RCI and RCR (17.24 ± 3.37 kg/m^2^ versus 18.17 ± 4.07 kg/m^2^). Alkerwi et al. indicate that the odds of overweight or obesity were significantly higher among immigrants as compared with a native population [40]. Higher openness to change in immigrants is correlated to increased probability of interactions involving discrimination, making the children more vulnerable to stress and consequently to an increase in food intake [41,42]. One hypothesis could be the compensation between those two processes, namely, decrease in BMI due to the adherence to MD but an increased caloric intake due to the adaptation stress. Moreover, no correlation between the place of residence, immigration period, and KIDMED index was observed in our data, neither in RCI nor in RCR population.

As Petre pointed out, Romania has a relatively high consumption of olive products. Compared with other countries, in Romania, more olives are consumed than in Poland and, surprisingly, than in Greece. Additionally, from 2001 to 2017, the imported quantity of olives increased by 84%. These data support our results that indicate that 39.12% of RCR consume olive oil, even though Romanian climate is inadequate for olive culture. There are several possible explanations, including the impact of nutrition education over food habits (especially in some social segments) and the practice of religious fasting periods associated with abstinence from animal foods [43]. 

One of the more significant findings which emerged from this study is that the high KIDMED index among RCI was significantly higher than the same index among RCR (68.09 versus 17.76, *p* < 0.05). Therefore, the immigration period (more than 5 years in 53.44% of cases) was long enough to produce a shift in the alimentation pattern. The importance of EVOO for health is increasingly well known, and greater accessibility to this product for RCI could be the key factor in improving their health [44]. RCR are less influenced by MD for cultural reasons linked also to the social-geographical distance with respect to the Mediterranean culture. Our data on adherence to MD among RCI are in line with those of other studies conducted in southern European countries: about half of pediatric individuals have an average adherence, while nearly half may have poor adherence. MD adherence varies in the literature from 4.3% in Greek 10–12-year-old adolescents to 53.9% in Spanish children [45,46].

The beneficial effects of MD must be supported by sustained physical activity both in the family and at school. Our data regarding physical activity are in accordance with literature data that indicate decreased physical exercise among immigrants: 33.62% of RCI declared no possibility for these activities, whereas 43.71% of RCR exercised more than 3 h per day. These results are not likely to be related to the place of residence as both groups exhibit quite a balanced place of residence distribution. The main difference we saw was in the percentage living in countryside (0.86% in RCI versus 8.45% in RCR) [47,48]. 

Bawaked et al. [49] observed that a higher level of schooling correlates with better adherence to MD in Spanish children compared with peers whose mothers have only primary education. A high maternal educational level implies better nutritional knowledge, food choices, and parenting practices. Full maternal awareness of healthy eating promotion appeared to have a protective effect against the development of childhood obesity and/or overweight. Most of the mothers of the RCI were unemployed (39.65%), while fathers were predominantly factory workers (54.31%), whereas the larger part of RCR parents were employees. These discrepancies shaped the time spent with children, the family earnings, and the dietary habits. The role of the mother in the family context is crucial in determining quality and quantity of children’s nutrition; the urbanization of life, linked to a more stressful lifestyle in migrant people, with less time spent on cooking, more time out of home, and dinner as the principal meal consumed with the family, could influence the food choices of the youngest children [26].

Adherence to the Mediterranean diet, as observed in the study by Albuquerque et al. [50], also depends on family income: higher adherence to MD is associated with higher cost. A possible consequence of augmented costs of healthier diets, such as MD, is an elevated prevalence of low-quality meals among less fortunate families, who cannot afford these foods, and therefore are more likely to buy energy dense foods (EDF), which are less nutrient-rich but lower in price. 

A number of caveats need to be noted regarding the present study. Adherence indexes, such as the KIDMED score, have been validated and used in epidemiological surveys, but their reliability and reproducibility in assessing diet quality in the individual subjects have not been demonstrated yet [45,46]. On the other hand, the KIDMED score is the most used index of adherence in pediatric literature. It is easy to complete by the respondent and easy to evaluate by the interviewer. 

Another limitation of the study is the difference in sample size between the two groups examined, caused by the challenges of recruiting subjects in Italy. This renders problematic the stratification of our data in order to obtain more specific results, related to age, immigration period, area of residence, etc. 

Another important assessment that should be taken into consideration in future research is the information about sleep pattern, level of parental education and income, sedentary habits (screen time), and school performance. 

We consider that public school plays a major role in the health education. It would be beneficial if the school assigned some curriculum time to health education and to training teachers to use classroom content and methods in order to promote health. A school dietician that could orient children towards a healthy alimentation and train their families to improve the choice of the primary alimentary products might be an important figure. The enlarged team involved in nutrition education should also include a pediatrician and a nutritionist. A two-point increase in the adherence to MD was significantly associated with a reduced risk in overall mortality (9%), mortality from cardiovascular diseases (9%), incidence of or mortality from cancer (6%), and incidence of Parkinson’s disease and Alzheimer’s disease (13%) [51].

## 5. Conclusions

In recent years, there has been an increase in the attention paid to nutrition, especially in lifestyle interventions for diseases prevention. Special attention was paid to eating behaviors due to the alarming boost in obesity rate, which is an increasingly common condition even among children. Learning to eat well is an important aspect of the child’s education and is likely to be conditioned by both the economic possibilities and the cultural level of the family of origin. 

Our study showed that the adherence to MD was higher among RCI. Besides cultural eating habits, in Italy, the access to olive oil and EVOO, fish, and legumes is easier. At the same time, RCI eat more junk food due to the influence of the westernized lifestyle.

These types of studies are increasingly important for assessing changes in eating habits in a globalized world, where cultural exchange and population movements are more and more frequent.

## Figures and Tables

**Figure 1 foods-10-02045-f001:**
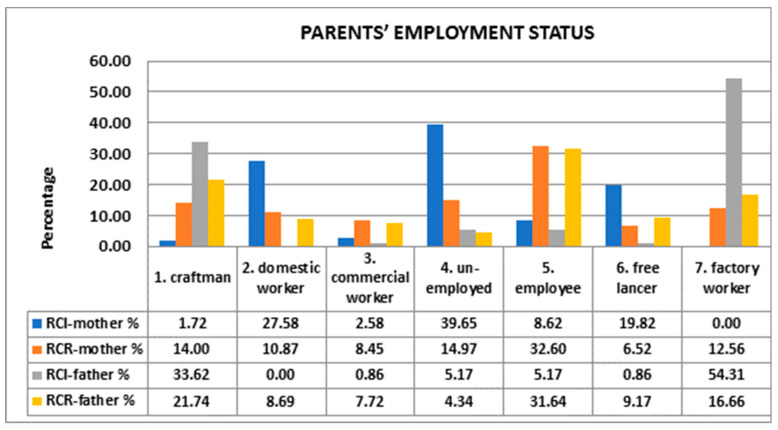
Comparison between parents’ employment statuses.

**Figure 2 foods-10-02045-f002:**
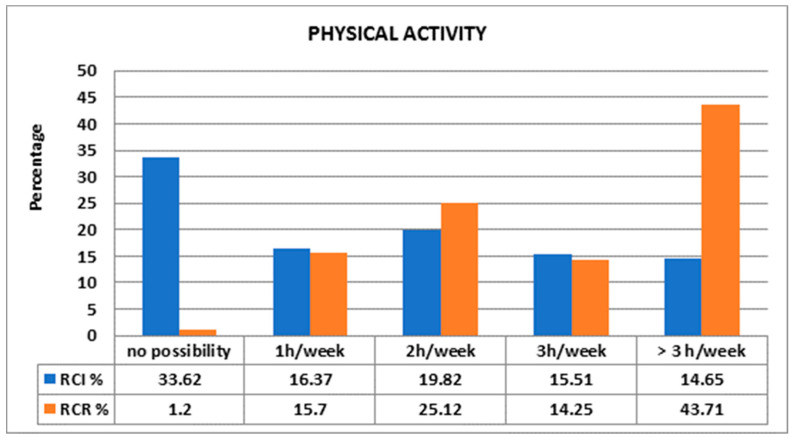
Comparison between physical activities of children.

**Figure 3 foods-10-02045-f003:**
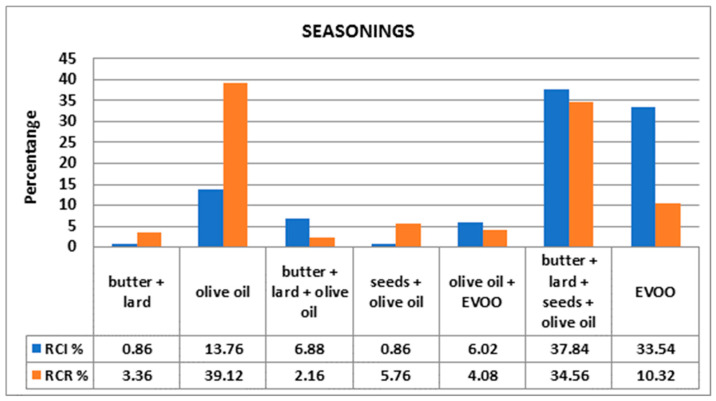
Comparison between preferred types of seasonings.

**Figure 4 foods-10-02045-f004:**
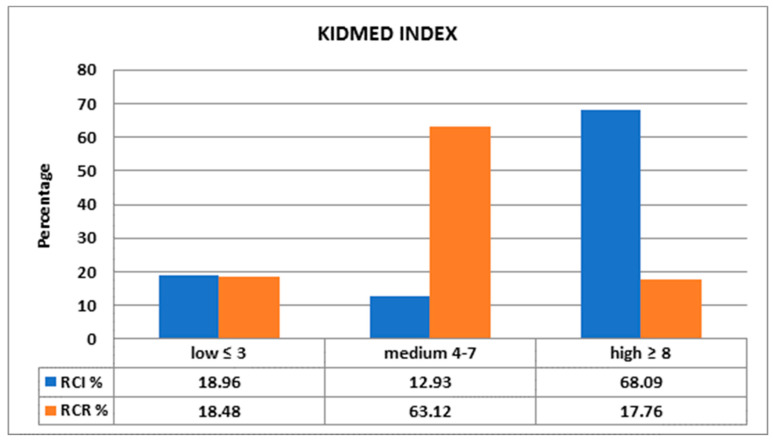
Comparison between KIDMED indexes.

**Figure 5 foods-10-02045-f005:**
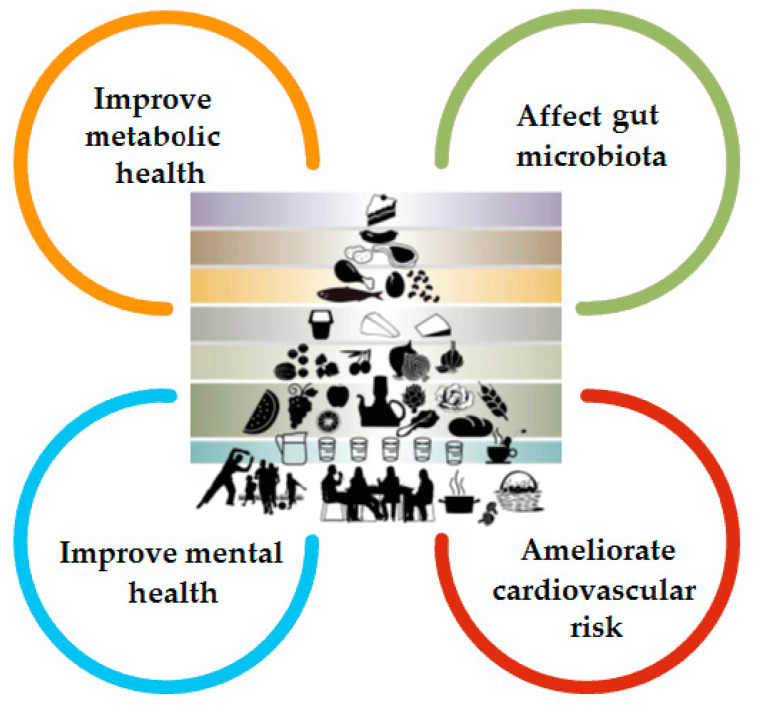
The benefits of MD (adapted after Bach-Faig et al.) [25].

**Table 1 foods-10-02045-t001:** KIDMED test, modified and adapted for this study from Serra-Majem [20].

KIDMED TEST	
Insert a value of 0 if the parent does not reply. For FRUIT and VEGETABLES consider only the question A or B.	
(A) Does the child eat one piece of fresh FRUIT every day, or drink a fresh fruit juice?	+1
(B) Does the child eat at least two pieces of fresh FRUIT every day?	+1
(A) Does the child eat once a day raw or cooked VEGETABLES?	+1
(B) Does the child eat more than once a day raw or cooked VEGETABLES?	+1
Does the child eat FISH (fresh or frozen) often (at least twice or three times a week)?	+1
Do you go with him/her to fast-food restaurants more than once a week (and the child eats hamburgers, French fries and drinks sweetened beverages)?	−1
Does he/she eat LEGUMES more than once a week?	+1
Does he/she eat PASTA AND RICE more or less every day? (five or more times a week)	+1
Does he/she eat BREAD OR CEREALS (without added sugar) for breakfast?	+1
Does he/she eat NUTS OR DRY FRUIT regularly? (at least twice or three times a week)	+1
Do you use OLIVE OIL at home?	+1
Does the child SKIP BREAKFAST in the morning?	−1
Does he/she eat milk derivatives, MILK, YOGHURT, or similar products, for breakfast?	+1
Does he/she eat BAKERY PRODUCTS OR INDUSTRIAL SNACKS for breakfast?	−1
Does he/she eat TWO YOGHURTS AND/OR CHEESE (maximum 40 g) during the day?	+1
Does he/she eat SWEETS AND CANDIES often during the day?	−1
KIDMED INDEX Low ≤ 3, Medium 4–7, High ≥ 8	TOTAL …

**Table 2 foods-10-02045-t002:** Demographic data.

Features	Participants (*n* = 530)	
	RCR (*n* = 414)	RCI (*n* = 116)
Sex: female	217	57
Sex: male	197	59
Body Weight (kg)	36.5 ± 15.36	26.629 ± 14.178
Body Height (m)	1.4 ± 0.21	1.198 ± 0.242
Body—Mass Index (BMI) (kg/m^2^)	18.17 ± 4.07	17.244 ± 3.371
Aged 1–5	18.59%	41.37%
Aged 6–9	32.12%	35.34%
Aged 10–12	28.26%	16.37%
Aged 13–16	21.01%	6.89%

RCR—Romanian children in Romania, RCI—Romanian children in Italy.

**Table 3 foods-10-02045-t003:** Comparison of the KIDMED items between RCI and RCR.

Type of Diet	KIDMED Score % RCI	KIDMED Score % RCR	*p*
Consumption of a fruit or a fruit juice every day	56.89	61.59	0.81
Consumption of a second fruit every day	35.34	29.47	0.015 *
Consumption of raw or cooked vegetables 1 time a day	56.89	58.70	0.72
Consumption of raw or cooked vegetables >1 time a day	37.93	23.67	0.0001 *
Consumption of fish regularly (at least 2–3 times a week)	61.21	31.88	0.0001 *
Eating >1 time per week at a fast food (hamburger) restaurant	44.82	15.94	0.0001 *
Consumption of beans >1 time per week	68.10	50.24	0.073
Consumption of pasta or rice almost every day (≥5 times a week)	95.68	33.33	0.0001 *
Consumption of cereals or grains (bread, etc.) for breakfast	87.93	83.09	0.71
Consumption of nuts regularly (at least 2–3 times per week)	31.03	57.97	0.0001 *
Consumption of olive oil at home	94.82	81.88	0.001 *
Skipping breakfast	25.86	22.95	0.59
Consumption of a dairy product for breakfast (yoghurts, milk, etc.)	87.93	90.82	0.80
Consumption of commercially baked goods or pastries for breakfast	75.00	39.13	0.0001 *
Consumption of yoghurts and/or cheese (40 g) daily	37.06	49.52	0.023 *
Consumption of sweets or candy several times every day	56.89	35.02	0.0001 *

* *p* < 0.05

## Data Availability

The study did not report any data.

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
