# Peer review of "Comparative Study Regarding the Adherence to the Mediterranean Diet and the Eating Habits of Two Groups—The Romanian Children and Adolescents Living in Nord-West of Romania and Their Romanian Counterparts Living in Italy"

_foods, 2021, doi:10.3390/foods10092045_

Round 1
Reviewer 1 Report
It is advisable to incorporate (for instance in a panel of the last figure) a pictorial or cartoon representation of the main results of the study to increase the overall impact of the manuscript.
The manuscript could benefit from a clearer exposition and a more focused discussion; moreover, the discussion fails to interpret the data in the context of what is known in the field: it sounds somehow redundant, as it largely summarizes again data already presented in the Results without placing them in the proper scientific context.
The following pertinent reports should be discussed:
PMID: 22633793
PMID: 34371970
PMID: 31771147
PMID: 28800091
PMID: 34204683
PMID: 34301736
PMID: 34204057
Author Response
Dear Sir / Madam
We want to thank you for your suggestions. We have considered the recommended bibliography and have included in our article a section on the benefits of MD. With all due respect, our main objective is to evaluate the adherence to MD among Romanian children, following and explaining certain parameters for a specific and complex situation, so that the suggested reports (which contain especially data on active compounds, mechanisms or health impact ) were very welcome in this new part. We supplemented the article with an image of the benefits of MD to increase the impact.

Reviewer 2 Report
The manuscript entitled ‘Comparative Study Regarding the Adherence to the Mediterranean Diet and the Eating habits of two groups- the Romanian children and adolescents living in Nord-West of Romania and their Romanian counterparts living in Italy’ presents interesting issue, however some corrections are needed
- Lines 54-55 – ‘Furthermore, MD has been associated with greater weight loss as compared to some other diets [3,4].’ – please reformulated, due to the fact, that not all diets are weight reduce diet. Individuals with proper body mass on MD should not reduce their weight but they should maintenance.
- Line 59 – ‘MD includes (…) low to moderate intake of fish’ – in comparison of other diet it is quite high intake (please refer to the WHO recommendation)
- Line 77 – ‘the influence of social media [12],1 education,’ – typos – please correct it
- Line 82 – ‘21,0 %’ it should be ’21.0 %’
- Lines 100-101 - BMI was calculated as 100 the ratio between weight (kg) and squared height (m2).’ - BMI of the children should be presented as BMI percentile (for age)!!! Not as BMI for adults. Please used WHO BMI percentile or national percentile (if available)
- Lines 103-104 – ‘we used an adapted version 103 of the KIDMED Test’ - what is the original language of the questionnaire. Was the questionnaire translated? Who did so? Any validation of the translated questionnaire? More information is needed about the validity and reliability of each measure. Additionally, any limitations in reliability and validity need to be addressed in the discussion.
- Line 120 – ‘Results were expressed as means±standard deviation’ - Was the normality of distribution tested? The information about it should be added and authors should be consequent. If data have normal distribution, they should be treated as such, if not, nonparametric tests should be applied. Please specify it.
- The table must stand alone (should be understandable without referring to the text – including abbreviations which should be explained as footnotes)
- Table 2 – please use dots instead of comas (e.g. ‘36,5 ± 15,36’ – it should be ’36.5 ± 15.4’) – one decimal point for weight and height is enough.
- Taking into account the % of age of children and adolescents it must be notice, that the RCR and RCI are not comparable! This is a serious bias!
- Figure 1 - please use dots instead of comas for values – please correct here and anywhere else
- Line 152 – ‘33,62%’ it should be ’33.62%’ – please correct here and anywhere else
- Table 3 - one decimal point for is enough – please correct here and anywhere else
- Line 200 –‘Some previous studies had found a correlation between MD and lower BMI [30-33].’ – please indicated only studied involving children and adolescents
- Line 203-204 – ‘In our research there is no difference between BMI of RCI and RCR (17,24 ± 3,37 Kg/m2 versus 18,17 ± 4,07 Kg/m2).’ – please recalculated the data on BMI percentile (especially if the age of children in sub-group is so differ)
- Line 213 – ‘As Peter L. points…” – please correct the references style (I can’t find this reference)
- Were both mothers and fathers of children and adolescents Romanian? If not – it biases - this aspect must be indicated.
Author Response
Dear Sir / Madam
We want to thank you for your suggestions. We want to thank you for your suggestions. We have made the requested changes and we hope that we have responded pertinently to all your comments. Specifically, we corrected all the minor errors (replacing comas or typewriting errors)
For “Lines 54-55” – we modified placing as moderate the intake of fish, and poultry meat and dairy products (milk, yoghurt and cheese) (≥ 2 servings per week)
For “Lines 100-101 and 203-204” regarding the use of BMI value, we explained also in text: Our study did not aim to assess adherence to MD based on weight status based of the children. BMI can be an important indicator of healthy growth and development and allows comparision between children of the same sex and age.
For “Lines 103-104” – th test is common use by other autors to test the adherence to MD. It was developed and validated by Serra-Majem et al [19]. We translated the questionare. The translation was made in Romanian and Italian by the authors of the study.
For “Line 120” – all the values were normally distributed. We clarified in text
We competed table 2 with explained abreviations
For “Line 200” - we replaced some of the studies.
For “Line 213” – we corrected the reference
Both of the parents were ethnic Romanian

Round 2
Reviewer 1 Report
-